# IMPLICIT CHAIN OF THOUGHT REASONING VIA KNOWLEDGE DISTILLATION

## ABSTRACT

To augment language models with the ability to reason, researchers usually prompt or finetune them to produce chain of thought reasoning steps before producing the final answer. However, although people use natural language to reason effectively, it may be that LMs could reason more effectively with some intermediate computation that is not in natural language. In this work, we explore an alternative reasoning approach: instead of explicitly producing the chain of thought reasoning steps, we use the language model's internal hidden states to perform implicit reasoning. The implicit reasoning steps are distilled from a teacher model trained on explicit chain-of-thought reasoning, and instead of doing reasoning "horizontally" by producing intermediate words one-by-one, we distill it such that the reasoning happens "vertically" among the hidden states in different layers. We conduct experiments on a multi-digit multiplication task and a grade school math problem dataset and find that this approach is able to outperform baselines that directly produce the answer by a large margin.

## 1 INTRODUCTION

Large language models have demonstrated significant capabilities in tasks that demand both language understanding and reasoning, such as multi-hop question answering (Yang et al., 2018; Yao et al., 2023b) and solving math problems (Hendrycks et al., 2021; Cobbe et al., 2021; Welleck et al., 2022; Wei et al., 2022b; Kojima et al., 2022; Chen et al., 2022; Yue et al., 2023; Chern et al., 2023). To elicit their reasoning abilities, a prevalent paradigm has been the chain-of-thought reasoning approach (Nye et al., 2021; Wei et al., 2022b; Kojima et al., 2022). Under this paradigm, models are trained or prompted to articulate intermediate steps before producing the final answer.

Although this approach aligns with human problem-solving strategies, it might not fully leverage the computational potential of these language models. Consider the transformer architecture (Vaswani et al., 2017), which can manifest computation both "horizontally" by generating words in sequence and "vertically" by processing through its many layers of internal hidden states. With models like GPT-4 having as many as 120 layers (OpenAI, 2023), one might wonder: Why not let these models reason internally, "vertically" through their layers, and present the solution without necessarily articulating every intermediate step? Such an approach would not only save the significant time cost of autoregressively generating the chain-of-thought: it may also allow models to develop more efficient, if less human-interpretable, methods of reasoning, unconstrained by human conventions.

While chain-of-thought (CoT) methods have achieved impressive successes, generating the CoT itself delays the production of the desired ultimate answer, and it is worth investigating whether insights from CoT methods can be exploited in models that directly produce the answer. Drawing inspiration from how the human brain compiles (Anderson, 2005) explicit, conscious, deliberate reasoning (System 2) to more implicit, automatic, intuitive thinking (System 1) (Kahneman, 2011), we seek a method to compile explicit CoT reasoning into a model that directly produces the final answer. We call this the *implicit chain-of-thought* approach. We take the internal states across transformer layers produced when generating the CoT in a model trained to do so, and train a model to predict a compressed encoding of this vertical sequence of states: the predicted sequence is then used as additional information at inference time for another model that directly generates only the final answer. In this sense, we compile the internal states that would be autoregressively generated horizontally in an explicit CoT model into the predicted vertical sequence of internal states which

is used to generate the answer directly. This replaces (horizontal) reasoning across an explicit CoT with implicit (vertical) reasoning from layer to layer.

Standard CoT training uses 'teacher forcing' to require the model to explicitly generate the CoT, but the new method uses a teacher model (which explicitly generates the CoT) to train another model to predict the teacher's internal states when generating the CoT: 'teacher teaching' rather than teacher forcing.

With this as our base, we propose a three-step strategy:

1. Mind-Reading the Teacher: We train a student model to "read" the teacher's "thought process" — the continuous hidden states during intermediate reasoning step generation. The student model, rather than replicating these steps, uses some of the teacher's hidden states to produce the answer.
2. Thought Emulation: We then employ knowledge distillation (Hinton et al., 2015; Kim & Rush, 2016) to train an emulator that predicts the teacher's hidden states from the input "vertically", across layers, eliminating the need for "horizontal" explicit reasoning steps.
3. Couple and Optimize: Finally, we combine the emulator, which predicts the teacher's thought process, with the mind-reading student, which produces the final answer from the emulated teacher's thought process. This combined system is then optimized end-to-end, allowing the student model to develop its own reasoning methods that might differ from the teacher's approach.

Our experiments show the potential of implicit chain-of-thought reasoning. On a synthetic multi-digit multiplication task, we found that while standard training cannot yield the final answer without explicit reasoning (even GPT-4 struggles with five-digit by five-digit multiplication), our method, applied to a GPT-2 Medium model, is able to provide direct answers for up to five-digit by five-digit multiplications. Moreover, when dealing with real-world tasks like grade school math problems, our method achieves a 22% accuracy on GSM8k (Cobbe et al., 2021) without the need for explicitly generating the intermediate steps.

The contributions of our work are as follows: First, we show the benefits of shifting from teacher-forcing to teacher-teaching by enabling faster generation. Second, we show the effectiveness of distilling explicit reasoning in a teacher to implicit reasoning in a student. Third, we demonstrate the improved performance on directly generating responses to math problems that results from chaining together the first two contributions. Our code, data, and pretrained models are available at `https://anonymous.4open.science/r/implicit_chain_of_thought/`.

## 2 Explicit, Implicit, and No Chain-of-Thought Reasoning

Consider a task that requires multi-step reasoning. Let $x$ be the input, $z$ the intermediate reasoning steps, and $y$ the output. As an example, for the multiplication problem $12 \times 3 = ?$, $x$ is $12 \times 3$, $z$ might be $6 + 30$ (making explicit the intermediate partial products), and $y$ is 36. The objective of a model trained for this task is to determine the conditional distribution $P(y|x)$. We distinguish three approaches for solving this task: no chain-of-thought reasoning, explicit chain-of-thought reasoning, and implicit chain-of-thought reasoning.

### 2.1 No Chain-of-Thought Reasoning

In this approach, models are trained to generate the final output $y$ using the input $x$, with the intermediate steps $z$ left out. Mathematically, we directly parameterize the mapping from input to output using a model $P_\theta(y|x)$ and train it with input-output pairs $(x, y)$. Using the $12 \times 3$ multiplication example, the model directly infers the answer 36, as illustrated in the "No CoT" column of Table 1. This method can work well for simple tasks, but expecting models to deduce answers for more complex tasks without intermediary guidance can be daunting, analogous to teaching multi-digit multiplication to students without showing the intermediate calculations.

### 2.2 Explicit Chain-of-Thought Reasoning

In explicit chain-of-thought reasoning (Nye et al., 2021; Wei et al., 2022b), models are trained to produce the intermediate steps $z$ before the final output $y$. Instead of only modeling $P(y|x)$, the model looks at the joint distribution $P(y, z|x)$ and breaks it down to $P_\theta(z|x)P_\theta(y|x, z)$. During

Table 1: Comparison between three methods of reasoning: no chain-of-thought (No CoT), explicit chain-of-thought (Explicit CoT), and implicit chain-of-thought (Implicit CoT). The reasoning process is illustrated using a multi-digit multiplication example: $12 \times 3 = 6 + 30 = 36$. Here, $x$ denotes the input '$12 \times 3 =$', $y$ denotes the output 36, and $z$ denotes the intermediate steps $6 + 30$. For the model $P_\theta$, observed variables are shaded. In No CoT, the model is trained to predict the output directly from the input. Explicit CoT predicts the intermediate steps before the final output. Implicit CoT is trained to reason internally using its hidden states and subsequently predict the output.

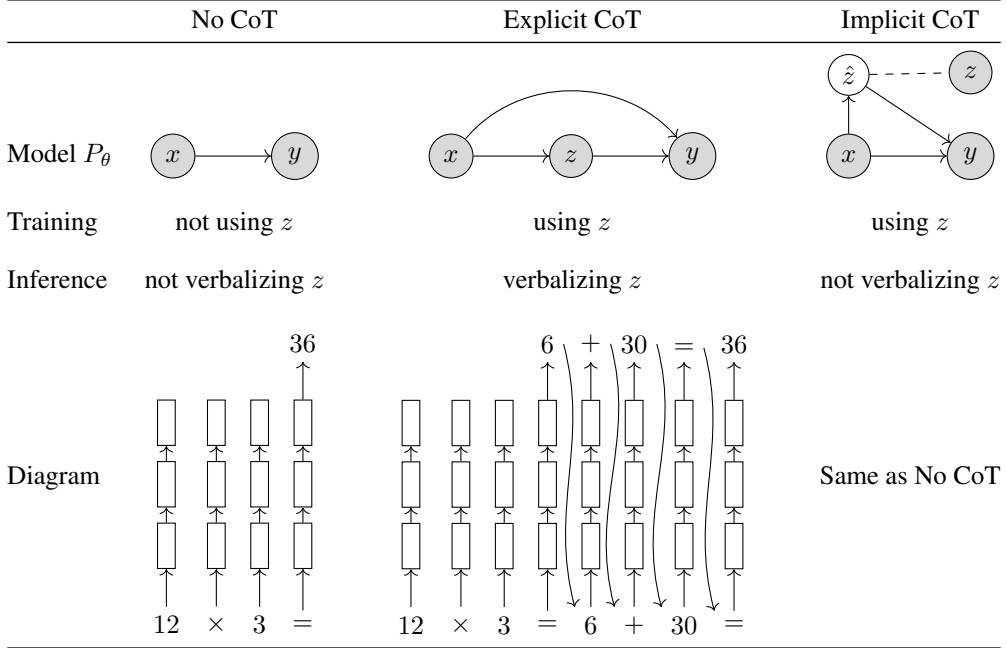

training, both components $P_\theta(z|x)$ and $P_\theta(y|x, z)$ are trained in a supervised way. At test time, the model first predicts the reasoning steps from the input, then the final output. For the multiplication $12 \times 3$, the model predicts $6 + 30$ first and then 36, as shown in the "Explicit CoT" column of Table 1. While this method breaks down the task into simpler steps, it can be verbose; as we'll see in later experiments, even a five-digit multiplication needs to generate seventy intermediate tokens.

### 2.3 Implicit Chain-of-Thought Reasoning

Implicit chain-of-thought reasoning is a middle ground between the two methods above. During training, the model sees intermediate steps $z$, but during testing, it doesn't explicitly produce them. Instead, it processes these steps in its internal states, labeled as $\hat{z}$, to produce the final output $y$. Formally, $P(y|x) \approx \int_{\hat{z}} P_\theta(\hat{z}|x) P_\theta(y|x, \hat{z})$. This mirrors how humans, once they've internalized a concept thoroughly, often bypass explicit reasoning, directly leaping to conclusions. Referring back to our multiplication example, the model directly predicts 36 for $x = 12 \times 3$, having computed the steps internally. The inference diagram for this is the same as the no chain-of-thought reasoning, as seen in Table 1 under "Implicit CoT".

### 3 Approach to Implicit Chain of Thought Reasoning

As a first step toward achieving implicit chain-of-thought reasoning, we outline a three-step strategy based on a teacher model trained for horizontal explicit chain-of-thought reasoning. First, we train a student model to generate the answer using the hidden states of the teacher model, which hold information about the intermediate reasoning steps. This allows the student to produce the final answer directly from the input and teacher states without needing the explicit reasoning steps. We call this "Mind-Reading the Teacher". Next, we apply knowledge distillation to train an emulator that can predict the teacher's hidden states from the input by reasoning vertically, a step we term

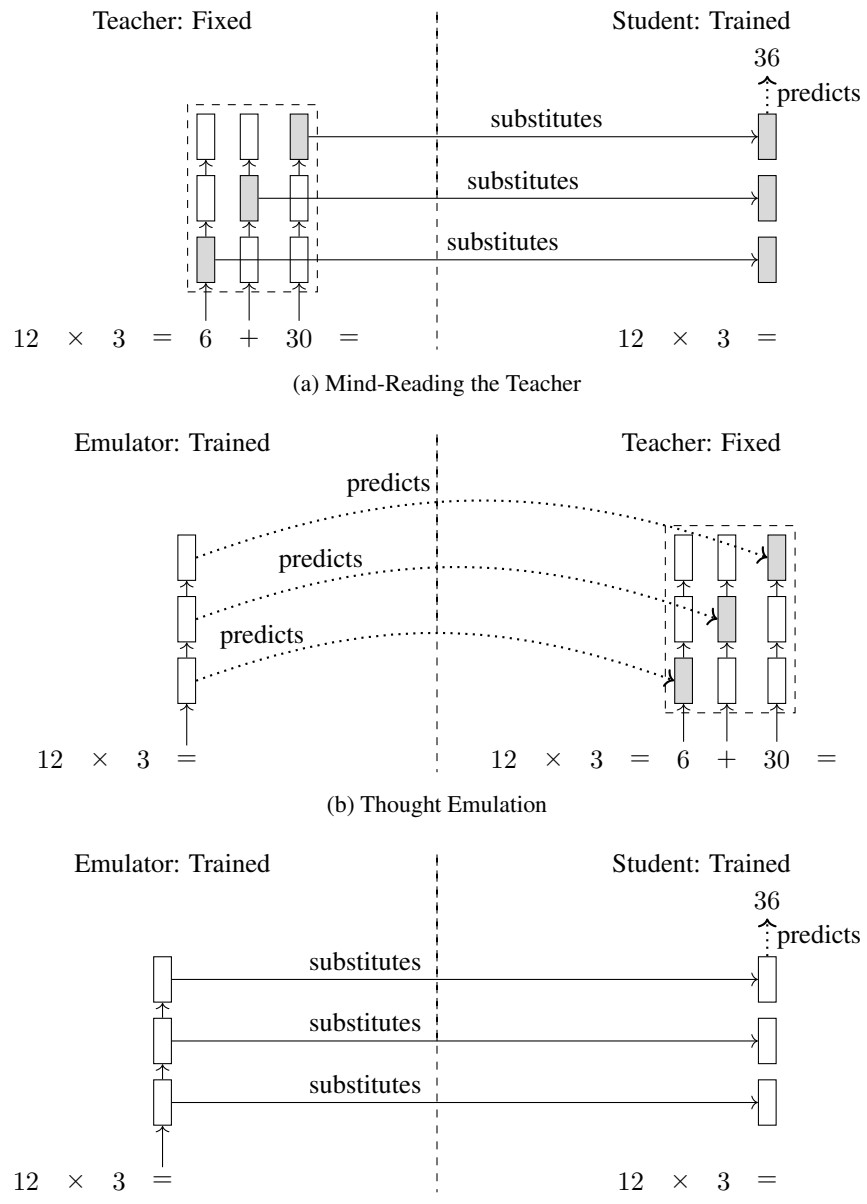

Figure 1: The three-step strategy for implicit chain-of-thought reasoning.. (a) Mind-Reading the Teacher: A student model "reads" the teacher's continuous hidden states (internal reasoning process) and uses them to produce the final solution. (b) Thought Emulation: An emulator is trained to predict the teacher's hidden states based on the given input, thus mimicking the internal reasoning process of the teacher without generating explicit horizontal reasoning steps. (c) Couple and Optimize: Integration of the emulator and the mind-reading student forms a combined system. This system is then finetuned end-to-end, enabling the student model's development of its own reasoning trajectories, possibly deviating from the teacher's method.

"Thought Emulation". Finally, we combine the student and emulator. Here, the student uses the teacher states predicted by the thought emulator to give the final output from the input. We then fine-tune this combined model end-to-end to improve the internal reasoning process, a step we name "Couple and Optimize". All these phases are illustrated in Figure 1.

Having outlined the general methodology, we now delve into the implementation specifics when employing the transformer architecture (Vaswani et al., 2017) for our teacher, student, and emulator models. Transformer's layer-wise processing offers a natural platform to leverage the vertical reasoning concept.

## 3.1 MIND-READING THE TEACHER

As the teacher processes the input, its intermediate reasoning steps, and the output, its hidden states capture token-related information. Specifically, for a transformer model with $L$ layers running on $T$ intermediate tokens, the hidden states can be expressed in a 2D matrix $\mathbf{z}$ of dimensions $L \times T$, with each element $\mathbf{z}_{lt}$ representing the hidden state at layer $l$ for intermediate token $t$.

**Information Extraction**  While the matrix holds $L \times T$ vectors, we only select $L$ vectors, allowing an emulator with an equal number of layers to predict just one vector per layer. Through experimentation, we found that simply taking the matrix's diagonal elements was effective. Assuming that the teacher model is autoregressive, the intuition is that predicting $\mathbf{z}_{11}$ is easy for the emulator since only one intermediate token is introduced[1]. Progressing diagonally, from $\mathbf{z}_{11}$ to $\mathbf{z}_{LL}$, we gradually add more intermediate tokens and layers, ensuring a gradient of increasing difficulty for the emulator, until $\mathbf{z}_{LL}$, which ideally has enough information to let the teacher start producing the output[2].

**Variable Length Chain-of-Thought**  Real-world scenarios may present variable number of intermediate tokens, resulting in a variable number of columns $T$. To handle this, we introduce a hyper-parameter $\Delta$ and take evenly-spaced columns while still selecting one vector per row. The selected $l$-th vector is $\mathbf{z}_{l,t_l}$, determined by:

$$t_l = \min(\lfloor 1.5 + \Delta(l-1) \rfloor, T).$$

In our experiments, we search over both fixed $\Delta$ values, and also a dynamic $\Delta$ value $\frac{T-1}{L-1}$ which adapts to the number of intermediate tokens $T$ in each example, based on validation performance.

**Student Training**  We use a student with the same number of layers as the teacher. Following the extraction of $L$ vectors, these vectors substitute the corresponding hidden states of the student right after input. Refer to Figure 1a for a visual illustration. The student model is then trained to predict the final answer, with the teacher model fixed.

## 3.2 THOUGHT EMULATION

At test time, the student cannot rely on the $L$ selected vectors $\mathbf{z}_1, \mathbf{z}_2, \ldots, \mathbf{z}_L$ (we abuse notation and omit the second index in $\mathbf{z}$) from the teacher, so we need to train an emulator to predict those $L$ vectors directly from the input. We use an emulator with the same number of layers as the teacher, such that after the input it only needs to predict one vector $\hat{\mathbf{z}}_l$ per layer, as shown in Figure 1b. We train this emulator by minimizing mean squared loss:

$$\min_{\hat{\mathbf{z}}_l} \sum_{l=1}^{L} \|\mathbf{z}_l - \hat{\mathbf{z}}_l\|_2^2. \tag{1}$$

**Multiple Reasoning Pathways**  When there exist multiple possible reasoning pathways, directly predicting the teacher's state using a mean squared loss would result in poor predictions, similar to using a single Gaussian distribution to fit a mixture-of-Gaussians distribution and only capturing the

---

[1]In our experiments the first intermediate token is always a special token separating input and reasoning.

[2]For predicting the first answer token, only the top teacher state is used; lower-layer states are only accessible to the second answer token and onward via attention. This is not an issue because in our experiments answers always start with a special token separating reasoning and output.

center. As an example, consider a grade school math problem: *Asumi has 30 books on history, 25 books on literature. How many books does Asumi have in total?*. The intermediate steps could be either (1) $30 + 25$ or (2) $25 + 30$, corresponding to two possible hidden states $\mathbf{z}_l^{(1)}$ and $\mathbf{z}_l^{(2)}$. The optimal solution using Equation (1) would be $\hat{\mathbf{z}}_l = (\mathbf{z}_l^{(1)} + \mathbf{z}_l^{(2)})/2$, which does not correspond to any valid reasoning path[3].

To account for multiple reasoning pathways, instead of predicting one $\hat{\mathbf{z}}_l$ per layer, we predict a *mixture* of components $P(\hat{\mathbf{z}}_l) = \sum_{c_l} P(\hat{\mathbf{z}}_l^{c_l}|c_l)P(c_l)$, such that each mixture component $c_l$ captures a different mode of the distribution of teacher states.

To parameterize this distribution, at layer $l$, assume the hidden state of the emulator is $\mathbf{h}_l$, we parameterize $P(\hat{\mathbf{z}}_l^{c_l}|c_l)$ as a Gaussian $\mathcal{N}(f(\mathbf{h}_l, c_l); 1)$ and the distribution over mixture components as $P(c_l) = g(\mathbf{h}_l, c_l)$.

Empirically, we found that directly fitting this mixture is prone to mode collapsing (He et al., 2019), where only a few mixture components get used. To alleviate this issue, we leverage the intermediate token $z_{t_l}$[4] at position $t_l$ and supervise $c_l$ to be the same as this token. The final objective is

$$\min_{\mathbf{h}_l} \sum_{l=1}^{L} \frac{\|\mathbf{z}_l - f(\mathbf{h}_l, c_l)\|_2^2}{2} - \log P(c_l = z_{t_l}). \qquad (2)$$

Taking the above example, at the first layer ($l = 1$ and $t_l = 1$), in case (1) we would supervise the mixture component $c_1$ to be "30" and fit $\hat{\mathbf{z}}^{30}$ to $\mathbf{z}_l^{(1)}$; in case (2) we would supervise $c_1$ to be "25" and fit $\hat{\mathbf{z}}^{25}$ to $\mathbf{z}_l^{(2)}$, hence the two cases are fit with different mixture components.

## 3.3 Couple and Optimize

We can now feed the emulator's predicted teacher states $\hat{\mathbf{z}}_l$ to the mind-reading student and optimize the entire system end-to-end by maximizing the probability of the final output. Importantly, as the combined system learns, the internal reasoning process might diverge from the teacher's approach. Note also that this step doesn't require training data that contains the intermediate reasoning steps.

For the mixture model, ideally we want to take the argmax of the reasoning pathway, but that operation is not fully differentiable. Instead, we approximate argmax using a softmax with low temperature, which is fully differentiable. See Appendix C for more details.

## 4 Experimental Setup

### 4.1 Data

We conduct experiments on two tasks: we first consider the multi-digit multiplication task from the BIG-bench benchmark (bench authors, 2023; Suzgun et al., 2023), which is the most challenging among arithmetic tasks (Yang et al., 2023). In particular, we use the four-digit ($4 \times 4$) and five-digit ($5 \times 5$) multiplication problems, since these two tasks prove very challenging to solve under no CoT. The second task we use is grade school math problems, which requires both language understanding and mathematical reasoning. In particular, we use the GSM8K dataset (Cobbe et al., 2021).

**Intermediate Steps** For the multiplication task, we break down the problem by multiplying the multiplicand by each digit of the multiplier and keep track of partial products and partial sums. On GSM8K, following Wei et al. (2022b) we use the natural language intermediate steps for explicit CoT. For training the teacher of implicit CoT, to minimize the gap between the number of transformer layers and the number of intermediate steps, we only keep the equations.

**Data Augmentation** In our preliminary experiments, we found that our proposed approach to implicit CoT requires a large training set, potentially due to its different mode of training compared to the pretrained language models we base on. Therefore, we generate synthetic data for both tasks.

---

[3]The emulator cannot distinguish the two cases because it only gets access to the input.
[4]we use unbolded $z$ to represent the intermediate tokens.

Table 2: Dataset Statistics. Sizes refer to the training set. The number of input, output, and intermediate tokens are median values on the validation set. The number of tokens are based on the GPT-2 tokenizer, and a special ending symbol is counted for both intermediate tokens and output tokens.

| Dataset | Orig Size | Aug Size | #Input Tokens | #Intermediate Tokens | #Output tokens |
|---|---|---|---|---|---|
| $4 \times 4$ Mult | 0 | 808k | 9 | 47 | 9 |
| $5 \times 5$ Mult | 0 | 808k | 11 | 75 | 11 |
| GSM8K-Aug | 7k | 378k | 51 | 59 | 2 |

For the multi-digit multiplication tasks, we randomly sample equations that do not overlap with the BIG-bench dataset. For GSM8K, we used GPT-4 (OpenAI, 2023) to generate 400k additional mathematical problems with the same format as GSM8K. We then clean the dataset and name it GSM8K-Aug. Note that for both tasks we do not change the original test sets. The statistics of the augmented datasets are shown in Table 2, which show that with explicit CoT the number of generated tokens rise by 5- to 30-fold. More data details can be found at Appendix A.

### 4.2 BASELINES

We compare our approach to both no CoT and explicit CoT. We compare to GPT-2 Small, GPT-2 Medium, GPT-2 Large, ChatGPT, and GPT-4. For smaller models from the GPT-2 family, we finetune them on the augmented training datasets. For ChatGPT and GPT-4, we use few-shot prompting to adapt them to the given task, whose details can be found at Appendix B.

### 4.3 MODELS

For implicit CoT, we finetune GPT-2 Small and GPT-2 Medium. For the diagonal teacher states, we normalize them to be zero-mean and standard deviation 1 to stabilize emulator training[5]. For the "Mind-Reading the Teacher" step, we add a trainable one-layer MLP on top of the teacher states before copying. For the "Thought Emulation" step, we add an LSTM network (Hochreiter & Schmidhuber, 1997) with self-attention to process the vertical hidden states before predicting the teacher states. For the mixture model, we add a linear projection on top of the emulator hidden states to predict the distribution over mixture components, and to compute $f(\mathbf{h}_l, c_l)$, we concatenate $\mathbf{h}_l$ with the embedding of the mixture component $c_l$, and then process them with a one-layer MLP.

We used the mixture approach for GSM8K-Aug but not for multiplication, because the multiplication intermediate steps are unique given any input by construction. For the mixture approach we use a temperature of 0.05 during "Couple and Optimize". See Appendix C for full model details.

## 5 RESULTS

Table 3 presents the main results. Compared to no CoT, our approach enables solving tasks previously not solvable without explicit CoT: for example, GPT-2 Medium only got 1.9% accuracy on $5 \times 5$ multiplication under the no CoT setting, whereas GPT-2 Medium got 96.4% accuracy under the implicit CoT setting. Interestingly, GPT-2 Small performed well on $4 \times 4$ multiplication, achieving 96.6%, while precipitously falling to 9.5% on $5 \times 5$, which might be due to the lack of sufficient layers to perform the necessary intermediate steps.

On GSM8K-Aug, implicit CoT enables directly producing the final answer with over 20% accuracy, whereas the best GPT-2 model with no CoT only achieves 17.0% accuracy. Surprisingly, GPT-4 with no CoT performs on par with GPT-2 Large finetuned with explicit CoT, which we suspect might be either due to data contamination or due to emergent capabilities at scale (Wei et al., 2022a).

Compared to explicit CoT, implicit CoT lags behind by a large margin, possibly due to two reasons: first, the base language models we used were all pretrained for horizontal reasoning; second, the number of layers we used in our experiments (24 for GPT-2 Medium) might not be sufficient for

---

[5]We found that higher layers tend to have hidden states of higher norms. We applied this normalization to each hidden vector, same as applying layer normalization (Ba et al., 2016) without trainable parameters.

Table 3: Main results. Accuracy measures exact match accuracy. Speed measures the number of examples per second during inference using a batch size of 1. [†]: few-shot prompting instead of finetuning, and Speed is measured based on API calls with a large variability across runs.

| Model | #Layers | $4 \times 4$ Mult | | $5 \times 5$ Mult | | GSM8K-Aug | |
|---|---|---|---|---|---|---|---|
| | | Accuracy | Speed | Accuracy | Speed | Accuracy | Speed |
| **No CoT** | | | | | | | |
| GPT-2 Small | 12 | 28.7% | 13.2 | 1.2% | 11.1 | 13.3% | 24.7 |
| GPT-2 Medium | 24 | 76.2% | 7.0 | 1.9% | 5.9 | 17.0% | 13.2 |
| GPT-2 Large | 36 | 33.6% | 4.8 | 0.9% | 4.0 | 12.7% | 9.1 |
| ChatGPT[†] | 96 | 2.2% | 1.0 | 0.0% | 1.4 | 28.1% | 1.8 |
| GPT-4[†] | - | 4.0% | 0.7 | 0.0% | 0.8 | 43.8% | 0.9 |
| **Implicit CoT** | | | | | | | |
| GPT-2 Small | 12 | 96.6% | 8.9 | 9.5% | 7.9 | 20.0% | 16.4 |
| GPT-2 Medium | 24 | 96.1% | 4.8 | 96.4% | 4.3 | 22.0% | 8.7 |
| **Explicit CoT** | | | | | | | |
| GPT-2 Small | 12 | 100.0% | 2.3 | 100.0% | 1.5 | 40.7% | 2.0 |
| GPT-2 Medium | 24 | 100.0% | 1.2 | 100.0% | 0.8 | 43.9% | 1.1 |
| GPT-2 Large | 36 | 100.0% | 0.8 | 99.3% | 0.6 | 44.8% | 0.7 |
| ChatGPT[†] | 96 | 42.8% | 0.1 | 4.5% | 0.1 | 61.5% | 0.2 |
| GPT-4[†] | - | 77.0% | 0.1 | 44.3% | 0.1 | 90.9% | 0.1 |

the number of reasoning steps needed. That being said, implicit CoT has a higher inference speed, especially for tasks with many intermediate steps such as GSM8K-Aug and $5 \times 5$ multiplication, since it's directly producing the final answer, with the only overhead being the emulator, which can also be parallelized in theory (although not in our experiments).

## 6 ANALYSIS

**Taking Different Subsets as Teacher's Thought Process**  In our main experiments, we took the diagonal elements from the matrix of teacher hidden states. Several other methods of extracting a compressed encoding of these hidden states did not perform as well. On the $4 \times 4$ multiplication task using GPT-2 Small, when we use diagonal, the validation accuracy is 100.0%; using first column gets 29.9%; using top row gets 84.4%; using bottom row gets 57.6%; using last column gets 58.7%.

**Mixture**  Due to the existence of multiple possible reasoning pathways, the mixture approach is crucial for GSM8K. Without the mixture approach, we achieve 11.2% validation accuracy on GSM8K-Aug (GPT-2 Small, $\Delta = 2$). With a mixture, this rises to 20.2%.

**Coupling & Optimization**  The "Optimize" part is important as well. On GSM8K-Aug with GPT-2 Medium and $\Delta = 1$, coupling the emulator and the mind-reading student without further optimization only results in a validation accuracy of 9.4%, compared to 22.0% after further optimization. Allowing the student to develop its own reasoning pathway is also important: if we fix the emulator and only optimize the student, accuracy drops to 13.0%.

For the mixture approach, since we supervised mixture components to be the same as the current intermediate token, we can map back the predicted mixture components to a string of words. Before the "optimize" step, these mapped words look similar to the intermediate reasoning steps produced by the teacher when $\Delta = 1$, and if we directly use them for prediction, we can get an accuracy of 9.4%. However, after the "optimize" step, the predicted mixture components are no longer interpretable, as shown in Table 4 in Appendix E,

## 7 RELATED WORK

**Emergent Capabilities**  Research has shown that, under sufficient optimization, language models can solve basic arithmetic tasks (Power et al., 2022). Even for tasks that require multi-step reasoning,

increasing both model and data size improves the model's direct performance. For example, Wei et al. (2022a) observed that test accuracy on the GSM8K dataset (no CoT) rises from below 5% to about 7% as the training FLOPs increase from $10^{21}$ to $10^{24}$. Concurrent to our work, Yang et al. (2023) trained a 2B language model to solve $5 \times 5$ multiplication with an accuracy of 89.9% through curriculum learning on 50M training examples. These findings demonstrate that sufficiently scaled models can internally reason over multiple steps. Our approach differs in its use of the teacher model's thought process to more efficiently attain these models.

**Knowledge Distillation**  Our "Thought Emulation" step is a type of knowledge distillation, where the teacher model transfers its knowledge to a student model (Hinton et al., 2015). Traditionally, this technique is used for model compression (Kim & Rush, 2016) or for non-autoregressive machine translation (Gu et al., 2018). In our approach, it's used to distill the teacher model's horizontal reasoning process into a vertical reasoning process in the emulator and the student model.

## 8   Limitations

**Lack of Transparency and Interpretability**  One of the main advantages of explicit CoT is its inherent transparency: the intermediate steps allow for easy interpretation of the model's reasoning process. In contrast, implicit CoT, by virtue of its internal processing within hidden states, lacks this transparency. While it achieves compactness and efficiency in generation, it sacrifices human interpretability, making it challenging to understand how the model arrives at its conclusions.

**Reliance on the Teacher's Thought Process**  Our current three-step strategy is, at a high level, trying to distill the teacher model's horizontal reasoning process into the vertical reasoning process of the student and the emulator. While the ultimate goal of implicit reasoning is to allow models to develop their own unique trajectories of reasoning, our initial approach still relies heavily on the teacher's thought processes for a starting point.

**Performance Discrepancies**  Our current results of implicit CoT still lag behind the performance of explicit CoT. However, this work is just a first step towards building implicit CoT, and there exists ample room for further optimization.

## 9   Conclusion and Future Work

In this work, we proposed the concept of implicit chain of thought reasoning for transformer-based language models, where reasoning is performed "vertically" among the transformer hidden states, instead of being performed "horizontally" in the form of generating intermediate tokens. This concept potentially enables the model to break away from the human-like reasoning process and develop its own internal reasoning process.

To operationalize this concept, we proposed a three-step approach—mind-reading the teacher, thought emulation, and coupling and optimization, where the high-level idea is to distill the knowledge of a teacher trained for horizontal reasoning into a student and an emulator trained for vertical reasoning. Experiments on an arithmetic multiplication task and a grade school math problem dataset show that, for the task of directly producing an answer, the proposed approach substantially improves the performance of transformer language models — although the task of explicitly producing a chain-of-thought improves the accuracy of final answers further, by a large margin.

We see many exciting future directions that can be built on top of this work. For example, instead of the three-step strategy, one might explore a fully end-to-end joint training strategy using a variational auto encoder (Kingma & Welling, 2022) by treating the model's internal reasoning process as an unobserved variable. Another direction would be using image modeling techniques such as diffusion (Sohl-Dickstein et al., 2015; Ho et al., 2020; Song et al., 2021) to train the thought emulator. One might also explore incorporating this approach into the pretraining process, such that a pretrained language model can both do horizontal explicit chain-of-thought reasoning, and also do vertical implicit chain-of-thought reasoning, as opposed to existing models whose performance gets much worse when not allowed to use explicit reasoning steps.

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

# A   DATA AUGMENTATION AND PROCESSING

## A.1   MULTI-DIGIT MULTIPLICATION

For each multi-digit multiplication task, we randomly sample two numbers and compute their product. We collected 808k training equations and 1k validation equations after removing duplicates. We use the BIG-bench data as test set. To generate the intermediate steps, we break down the problem by multiplying the multiplicand by each digit of the multiplier, and we keep track of both partial products and partial sums. To make the task easier, we reverse the order of the digits such that lower digits come first. For example, for $917 \times 412$, the intermediate steps are

$$4\ 3\ 8\ 1\ +\ 0\ 7\ 1\ 9\ 0\ (\ 4\ 0\ 0\ 1\ 1\ )\ +\ 0\ 0\ 8\ 6\ 6\ 3,$$

where we break down $917 \times 412$ into $917*2 + 917*10 + 917*400 = \underbrace{1834 + 09170}_{11004} + 366800$.

Note that the partial sum $11004$ is reversed and written in the parentheses as $4\ 0\ 0\ 1\ 1$.

## A.2   GRADE SCHOOL MATH PROBLEMS

We use the training set of GSM8K (Cobbe et al., 2021) as a seed dataset and generate similar examples by prompting GPT-4. We used the below prompt template and a temperature of 1 for diversity of the generated dataset:

> Create 5 new math word problems following the JSON format of the given examples.
>
> Example math word problems:
>
> 1): {"question": "Meena bakes 5 dozen cookies for the school's bake sale. She sells 2 dozen cookies to her biology teacher, Mr. Stone. Her friend Brock buys 7 cookies, and her friend Katy buys twice as many as Brock. How many cookies does Meena have left?", "answer": "Meena bakes a total of 5 x 12 = ⟨⟨5*12=60⟩⟩60 cookies. Mr. Stone buys 2 x 12 = ⟨⟨2*12=24⟩⟩24 cookies. Brock buys 7 cookies, so Katy buys 2 x 7 = ⟨⟨7*2=14⟩⟩5 14 cookies. Meena sells a total of 24 + 7 + 14 = ⟨⟨24+7+14=45⟩⟩45 cookies. She has 60 - 45 = ⟨⟨60-45=15⟩⟩15 cookies left. #### 15"}
>
> 2): [...]
>
> 3): [...]
>
> 4): [...]
>
> 5): [...]
>
> Similar examples:
>
> 6):

Each time we prompt GPT-4, we take 5 random examples uniformly sampled from the training set of GSM8K. We prompted GPT-4 80k times, resulting in 400k generated examples. We then filter out examples with invalid JSON format, or examples whose intermediate steps don't lead to the same final answer (for example, in the above example, the last equation is ⟨⟨60-45=15⟩⟩, which matches

the final answer 15, so it is valid). After applying this filtering, we got a dataset of 379k examples, where we leave 1k for validation. We use the same test set from GSM8K.

For training explicit CoT, we used the natural language intermediate steps, which was shown in Wei et al. (2022b) to perform better than using equations. For example, for the input *Tom bought his games for $200. They tripled in value and he then sold 40% of them. How much did he sell the games for?*, the intermediate steps are *The value of the games increased to 200\*3=$600 So he sold 600\*.4=$240 worth of games*. For training the implicit CoT teacher, we only use the equations as intermediate steps based on our finding that more intermediate steps generally need more layers for the implicit chain-of-thought approach. For the same input, the intermediate steps are *200\*3=600 600\*.4=240*.

## B  FEW-SHOT PROMPTING BASELINES

In Table 3, we used ChatGPT and GPT-4 as baselines. For these baselines, we use few-shot prompting with five examples and a temperature of 0 (greedy decoding). Each time we randomly sample five example demonstrations from the training set. For the arithmetic multiplication datasets, we used the original numbers instead of using the reversed digits, and removed the whitespaces between digits.

For the no CoT setting, below shows an example prompt for $4 \times 4$ multiplication:

> Answer the final question following the exact format of the given examples. Do not break the problem down, directly produce the answer.
> Example problems:
> Q: 5646 * 1576
> A: #### 08898096
> [...]
> Q: 7560 * 3228
> A: #### 24403680
> Question to answer:
> Q: 1668 * 4380

For explicit CoT, an example prompt for $4 \times 4$ multiplication is:

> Answer the final question following the exact format of the given examples. Do not output anything else.
> Example problems:
> Q: 5646 * 1576 A: 1): 6 * 5646 = 33876 (partial sum 0 + 33876 = 33876) 2): 70 * 5646 = 395220 (partial sum 33876 + 395220 = 429096) 3): 500 * 5646 = 2823000 (partial sum 429096 + 2823000 = 3252096) 4): 1000 * 5646 = 5646000 (partial sum 3252096 + 5646000 = 8898096) #### 8898096
> [...]
> Question to answer:
> Q: 1668 * 4380

## C  MODEL DETAILS

### C.1  MIND-READING THE TEACHER

In this step, for each layer, we add a trainable one-layer MLP on top of the teacher state before using it to substitute the corresponding student state. This MLP first uses a linear layer to project the teacher state of size $H$ to a vector of size $4H$, and then we apply ReLU before projecting it back with another linear layer to project it back to size $H$.

In experiments, we search over $\Delta$ values from $\{1, 2, 3\}$, and also the dynamic $\Delta$ value $\frac{T-1}{L-1}$. We found that the dynamic $\Delta$ value performs the best for the arithmetic tasks. For GSM8K, we found $\Delta = 2$ works the best for GPT-2 Small and $\Delta = 1$ works the best for GPT-2 Medium.

## C.2 THOUGHT EMULATION

We discuss the more general mixture approach first. At each layer $l$, we first compute the corresponding column in the intermediate steps $t_l$ based on $\Delta$. Denote the emulator hidden state at this layer as $\mathbf{h}_l$, then we parameterize the mixture distribution $P(c_l)$ using a linear projection to the size of the vocabulary (since we supervised $c_l$ to be the same as the corresponding word $z_{t_l}$ in the intermediate steps), and then we use a softmax to get a valid probability distribution. Then in order to parameterize $f(\mathbf{h}_l, c_l)$, we embed $c_l = z_{t_l}$ into a vector of size $H$ ($H$ is the same as the transformer's hidden size), and concatenate this vector with $\mathbf{h}_l$, and pass through a one-layer MLP. This MLP has the same architecture as the MLP described in "Mind-Reading the Teacher" but uses a separate set of parameters.

In the original transformer architecture (Vaswani et al., 2017), the hidden state at the current layer is directly used as input to the next layer. Here we cannot follow that formulation, as then it cannot account for which mixture component is being used (since $\mathbf{h}_l$ doesn't contain information about $c_l$. Therefore, we take $f(\mathbf{h}_l, c_l)$, which contains information about $c_l$, and use an LSTM (Hochreiter & Schmidhuber, 1997) with self-attention to process it, and take the output to feed to the next transformer layer. The LSTM with self-attention is implemented similar to Luong et al. (2015), where we first project $f(\mathbf{h}_{1:l}, c_{1:l})$ into keys and queries, and then we use $f(\mathbf{h}_l, c_l)$ as query to attend to $f(\mathbf{h}_{1:l-1}, c_{1:l-1})$ using dot attention, and compute a weighted sum of previous keys using attention weights. We then concatenate the resulting vector (typically termed the context vector) with the output of the RNN and use a linear projection to project it back to size $H$ as the output of the LSTM. This context vector is also added to $f(\mathbf{h}_{l+1}, c_{l+1})$ as the input to the next step of LSTM. Finally, we feed in the output of the LSTM to the next transformer layer $l + 1$.

When not using the mixture approach, we simply set $c_l$ to 1 in the above process.

## C.3 COUPLE AND OPTIMIZE

The couple and optimize step is straightforward, with the exception of using the mixture approach. Ideally we want to take the argmax of the predicted mixture component at each layer, corresponding to committing to the most likely token at each reasoning step, but argmax is not fully differentiable. To make the computation fully differentiable, inspired by Gumbel-Softmax (Jang et al., 2017; Maddison et al., 2017) we use softmax with temperature to temper the distribution over mixture components $c_l$:

$$P(c_l; \text{temperature}) \propto P(c_l)^{1/\text{temperature}}.$$

With this tempered distribution, we compute a weighted sum $\bar{\mathbf{c}}_l$ over the one-hot representation of $c_l$, and compute $f(\mathbf{h}_l, \bar{\mathbf{c}}_l)$, where when this function computes embeddings of $c_l$, it computes a weighted sum of all embeddings in the vocabulary using $P(c_l)^{1/\text{temperature}}$ as weights. This process is fully differentiable, and when the temperature goes to zero, we recover taking the argmax of $P(c_l)$. In our experiments, we fix the temperature to a small value 0.05.

For the arithmetic tasks, we finetune both the emulator and the student after coupling. But for GSM8K, we found the coupled model tends to overfit even with the augmented dataset, and fixing the student alleviates overfitting.

# D OPTIMIZATION DETAILS

We use AdamW to optimize all our models (Kingma & Ba, 2017; Loshchilov & Hutter, 2019) with a batch size of 32 and a learning rate of 5e-5. For $4 \times 4$ multiplication, we trained the baselines for 30 epochs and the student model in implicit CoT for 15 epochs. For $5 \times 5$ multiplication, we trained both the baselines and the student model in implicit CoT for 40 epochs. For GSM8K, we trained both the baselines and the student model in implicit CoT for 15 epochs. For thought emulation, we trained for 30 epochs. For couple and optimize, we trained for 10 epochs on $4 \times 4$ and $5 \times 5$ multiplication and 20 epochs for GSM8K.

Table 4: Visualizing the predicted mixture components. We use GPT-2 Medium with $\Delta = 1$, such that layer $l$'s mixture component was supervised to be the $l$-th token of the intermediate steps in the Thought Emulation step. Before couple and optimize, the task accuracy is 11.2% on the validation set and afterwards it rises to 22.0%. If we use the mapped mixture components to derive the final answer, before couple and optimization we get 9.4% and after it we get 0%.

| Ground Truth $z$ | Predicted Before Couple & Opt | Predicted After Couple & Opt |
|---|---|---|
| 4*2=8 8*4=32 40-32=8 | 4*2=8 4*8=32 32- initiated=14 | rewrite HELPonialrunnerGreek 6 inscribedidget Store diversion – Speedileen grasped victimized648 setup official delinqu "# lawful HELPatin |
| 10*2=20 10+20=30 | 10*2=20 10+20=30 | rewrite HELPonialrunnerGreek 6 inscribedidget Store opens – solderileen graspedAccording648 PharaohPosarry HELP untneath floors |
| 320+430=750 400+300=700 750+430+400+700=2280 | 320+230=550 340+440=780 300+310=780 384+960=RPG40 | rewrite HELPonialrunnerGreek Thankfully inscribedidget Store diversion – victimizedileen MO-TAccordinglectedileenPos delinqu creat Tamil Rai conceptual |
| 4/2=2 16/2=8 8*2=16 4*16=64 | 16/2=8 8*2=16 16/4=4 | rewrite HELPonialrunnerGreek Thankfully inscribedidget Store diversion calib solderileen grasped RakousseAcc victimized valuableper565 HELP/ |

## E   VISUALIZING THE INTERNAL REASONING PROCESS

Since we supervised the mixture components $c_l$ using the intermediate tokens $z_{t_l}$, when $\Delta = 1$ we can map back the mixture components with highest probability into words in the vocabulary and visualize the reasoning process. As shown in Table 4, right after the Thought Emulation step but before the Couple and Optimize step, the mixture components look quite similar to the intermediate steps in data (which is not surprising given that's how we trained them). However, after further optimization of the coupled system, the mixture components no longer align with human-interpretable reasoning steps, indicating that the model's internal reasoning process might differ from that of human's.

## F   MORE RELATED WORK

**Chain-of-Thought Reasoning**   To enable language models to handle tasks that require multi-step reasoning, researchers have advocated training these models to explicitly generate intermediate computation steps (Nye et al., 2021; Sakenis & Shieber, 2022). With the rise of large pretrained models, methods that do not require training these models have emerged. For instance, Wei et al. (2022b) introduced chain-of-thought prompting using few-shot examples with intermediate steps. Similarly, Kojima et al. (2022) guided models to "think step-by-step" in a zero-shot setting. Further research has explored alternative prompting data structures (Yao et al., 2023a; Long, 2023; Besta et al., 2023), optimal CoT prompting techniques (Wang et al., 2023; Fu et al., 2023a), applying CoT to generate programs (Chen et al., 2022), use APIs (Yao et al., 2023b; Schick et al., 2023), and even in vision domains (Gupta & Kembhavi, 2023). Yet, these methods all require explicit intermediate steps, while our work directly generates the final answer.

**Reinforcement Learning for NLP**   In our work, continuous hidden states of the model are utilized for reasoning. Since this system is entirely differentiable, gradient descent is employed for optimization. Another avenue could involve letting models form their own symbolic reasoning path-

ways, potentially distinct from human reasoning, and fine-tuning the system through reinforcement learning (Schulman et al., 2017; Stiennon et al., 2020; Caccia et al., 2020; Ramamurthy et al., 2023; Ouyang et al., 2022). One could design a reward system based on the accuracy of the final answer and the efficiency of the reasoning pathway, akin to auto-prompting approaches (Zhou et al., 2023b; Singh et al., 2023; Deng et al., 2022; Zou et al., 2023).

**Mathematical Reasoning using LMs**   Large LMs, when properly prompted or fine-tuned, can perform multi-step mathematical reasoning with high accuracy. For instance, Cobbe et al. (2021) and Hendrycks et al. (2021) evaluated LMs on various math datasets, showing their capabilities in solving problems ranging from basic arithmetic to advanced calculus and differential equations. Recent progress includes innovative approaches like problem decomposition (Zhou et al., 2023a), complexity-based prompting (Fu et al., 2023a), program-of-thoughts methods (Gao et al., 2023; Chen et al., 2023b), and reranking solutions with verifiers (Ni et al., 2023; Li et al., 2023). Additionally, the development and utilization of new datasets, such as those constructed in Yue et al. (2023), have further improved the mathematical reasoning capabilities of LMs through fine-tuning.

**Efficient Inference of Transformer LMs**   The transformer architecture, despite its effectiveness in various NLP tasks, poses challenges in terms of computational efficiency, particularly during inference. To address this, researchers have explored several approaches. Quantization, as demonstrated by Xiao et al. (2023), offers improved inference speed albeit with some trade-off in precision. Knowledge distillation, another strategy, involves training a smaller, more efficient student LM to emulate the performance of a larger teacher LM, as explored by Kim & Rush (2016). Speculative decoding, a technique employed by Leviathan et al. (2023) and Chen et al. (2023a), leverages a smaller LM to predict future tokens, which are then corrected in parallel using the larger LM. A novel approach, lookahead decoding, proposed by Fu et al. (2023b), integrates the Jacobi iteration method to predict and verify future n-grams, improving the throughput of transformer LMs during inference.

