# OpenReview forum: "Implicit Chain of Thought Reasoning via Knowledge Distillation"
_ICLR.cc/2024/Conference — Submitted to ICLR 2024_

### Official Review · Reviewer_t8BA · 2023-10-23

**Soundness:** 2 fair
**Presentation:** 3 good
**Contribution:** 2 fair
**Rating:** 6
**Confidence:** 4

**Summary:**

This paper proposes a method to reason "vertically" across layers by using a chain-of-thought (CoT) model's internal hidden states.
The method relies on a teacher model that is originally trained to predict intermediate reasoning steps in natural language before outputing the final answer. The idea is to take the internal states of the teacher model across layers produced when generating the CoT steps. Then an emulator network is trained to predict a pre-defined sequence from the $L\times T$ matrix of states, with $L$ being the number of layers and $T$ being the number of tokens in the CoT steps. That (vertical) sequence of states is then used to predict the final answer.

Concretely, the method consists of three steps:
1. A network is first trained to predict the final answer to a reasoning question given the question and $L$ hidden states from the $L$-layered teacher model.
2. An emulator model is trained to predict the teacher’s vertical hidden states (when generating the CoT tokens) from the input (distillation step).
3. Finetuning the combine system that links the emulator with the network in step 1


The method is tested on 4 and 5 -digit multiplication and GSM8k, and compared to various GPT2 baselines, chat-GPT and GPT-4.
The method has a stronger performance than models trained to predict the answer directly, but a weaker performance than models with explicit CoT reasoning. The method is however faster than explicit CoT reasoning models.

**Strengths:**

The proposed method is original and interesting. It tries to address a challenging task for language models, ie: multi-step reasoning

The paper is well written and clear to understand.

The experimental section contains interesting ablation studies that shows the importance of various components. It is good to see the effect of selecting different hidden states from the teacher network, the importance of mixture on GSM8k, and the importance of optimizing both the emulator and student network weights after coupling the two.

**Weaknesses:**

1. Given that the method requires a teacher model that does explicit CoT reasoning to distill into the emulator, it should be evaluated against these models. Unfortunately, the proposed method is weaker than explicit CoT models, although faster.
Overall, this method trades interpretability and performance (explicit CoT) for speed, and it doesn't seem like a good trade-off given the extensive literature on making faster inference Transformers.

2. Another weakness of the proposed approach is that it requires significantly more training data. This is potentially due to the much different way of training the student network compared to the pre-trained model as mentioned by the authors. Could a non-pretrained model be better at this task?

3. Eventually, the literature review section is very light. GSM8k has been a popular benchmark for many reasoning methods. The paper could benefit from further discussion on related methods to do multi-step reasoning. In addition, fast inference transformers is also an active research domain. Some discussion about the field should be added.

Comment:

- In Section 3.1, the $L \times T$ matrix of the teacher model comes from a Transformer architecture, which (by default) has an attention matrix over **all** previous layer tokens.
So the assumption that “_Progressing diagonally, from z11 to zLL, we gradually add more intermediate tokens and layers_” is not always true. For this assumption to be true, you also need to assume that the attention matrix of the teacher transformer is autoregressive, ie: conditional from left-to-right. Such clarification should be added. It is only clear that this is the case by looking at the architecture chosen (GPT-2) which is indeed auto-regressive.

**Questions:**

With the current selection mechanism of hidden states described in Section 3.1, if the number of layers is greater than the number of tokens (L=4, T=3, delta=0.66), then it may be the case that $t_l$ doesn’t reach the last token in the sequence ($t_l$ = [1, 1, 2, 2] with L=4, T=3). How often is this happening? Could you find a better selection formula?

In Section 4.1, the author mentions that “_For training the teacher of implicit CoT, to minimize the gap between the number of transformer layers and the number of intermediate steps, we only keep the equations_”. Did you try to include the original explanation? What was the impact on performance?

Overall, this work proposes one way to combine a $T\times L$ weight matrix into a vector of L weights. It seems like the authors also tried “first column”, “top row”, and “bottom row”. Why not “last column”? That seems to be the one containing the most information.

Did you try to train a model from scratch? Since the task is very different from pre-training, maybe the same performance can be obtained with less data on a network trained from-scratch?

---

> ### Author Response · Authors · 2023-11-23
>
> Thank you for your comments.
>
> ### Re: this method trades interpretability and performance (explicit CoT) for speed, and it doesn't seem like a good trade-off given the extensive literature on making faster inference Transformers
>
> Our method is orthogonal to faster inference methods such as quantization and pruning (although not speculative decoding). Also, efficiency is not our only goal — our deeper aim is to break from the notion that LMs must adhere to human-like reasoning. Instead, we hope to enable LMs to develop their own reasoning pathways, leveraging their capabilities quite different from those we provide them with.
>
> ### Re: Another weakness of the proposed approach is that it requires significantly more training data. This is potentially due to the much different way of training the student network compared to the pre-trained model as mentioned by the authors.
>
> We suspect that explicit CoT benefits more from pretraining than implicit CoT, since the pretraining data likely contains similar explicit CoT examples (which is supported by the experiment of training models from scratch below). That said, we used the same amount of training data for all experiments, so the comparison is still fair.
>
> ### Re: Did you try to train a model from scratch
>
> Thanks for the insights! We conducted experiments on $5\times5$ Mult where all models are trained from scratch. We found that the teachers (Explicit CoT) can still reach 100% accuracy. Interestingly, training from scratch improves the performance of implicit CoT: e.g., implicit CoT reaches 87.9% accuracy under no pretraining (using GPT-2 Small), much higher than with pretraining (9.5%). We suspect that pretraining for explicit CoT might position the models in a mode not amenable for implicit CoT reasoning. We have included the full results in the below table:
>
> |                     	| GPT-2 Small | GPT-2 Medium |
> |-------------------------|------------:|--------------|
> | **With Pretraining**	|         	|          	|
> | Explicit CoT        	| 100%    	| 100%     	|
> | Implicit CoT        	| 9.5%    	| 96.4%    	|
> | **Without Pretraining** |         	|          	|
> | Explicit CoT        	| 100%    	| 100%     	|
> | Implicit CoT        	| 87.9%   	| 98.2%    	|
>
> ### Re: Literature review is light
>
> We have added more related work on multi-step reasoning and efficient transformer inference methods in the revised draft (Appendix F).
>
> ### Re: For this assumption to be true, you also need to assume that the attention matrix of the teacher transformer is autoregressive
>
> We have clarified in the updated paper (Section 3.1 - Information Extraction) that we assume that the teacher is autoregressive when we discuss the intuition behind taking the diagonal elements.
>
> ### Re: it may be the case that doesn’t reach the last token in the sequence
>
> Thanks for attending to this detail! Under the dynamic $\Delta$ setting, the last position taken will always be T, since the last layer is $l=L$, and $\Delta=\frac{T-1}{L-1}$, so $t_l=1+\Delta(l-1)=1+T-1=T$. In your example, L=4, T=3, the positions before rounding would be [1, 1.67, 2.33, 3], so after rounding the last one will be 3 instead of 2.
>
> That said, we do have a typo in the paper: instead of taking the floor in computing $t_l$ (Section 3.1 - Variable Length Chain-of-Thought), we actually rounded it to the nearest integer in our implementation (see https://anonymous.4open.science/r/implicit_chain_of_thought/src/models/teacher.py line 48). We have fixed this typo by adding 0.5 such that taking the floor is the same as rounding.
>
> ### Re: Did you try to include the original explanation? What was the impact on performance?
>
> In response to your question, we conducted an experiment on GSM8K using GPT-2 Small with implicit CoT. We found that incorporating the full explanation into the model resulted in a reduced validation accuracy of 15.6%, compared to 20.2% when only equations were used. We suspect that equations provide reasoning information in a more consistent and uniform manner. In contrast, combining text with equations introduces a degree of non-uniformity, where the numerical and symbolic components of equations carry more substantive information relative to the accompanying textual words. This discrepancy in information density between text and equations might be a contributing factor to the observed difference in model performance.
>
> ### Re: Why not “last column”?
>
> Thanks for pointing this out. We have added the last column result in the updated paper. The result is 58.7%, which is worse than taking the diagonal (100%).

---

### Official Review · Reviewer_XfmE · 2023-11-01

**Soundness:** 2 fair
**Presentation:** 3 good
**Contribution:** 2 fair
**Rating:** 3
**Confidence:** 3

**Summary:**

The paper proposes implicit chain-of-thought reasoning, where a language model is trained to conduct CoT reasoning internally without consuming the context window. The proposed method first a student model to predict the output based on the selected hidden states from a teacher model that conducts CoT reasoning. Then they train an emulator to predict the hidden states to mimic the CoT reasoning process. Finally, both student and emulator models are coupled and trained from end to end so that during the inference time, the whole system can conduct implicit CoT reasoning.

**Strengths:**

1.	The proposed method distills the CoT reasoning capability from a large model to a small one. The small LM does not have to consume its context window to conduct CoT reasoning.
2.	The paper is well-organized and easy to follow.

**Weaknesses:**

1.	As indicated by the authors as well, such implicit CoT reasoning is not interpretable and it is hard to tell whether indeed the proposed system is conducting CoT reasoning or is simply learning some reasoning shortcuts.
2.	The proposed method may not generalize compositionally to questions requiring more reasoning steps or just out-of-distribution data. It forces the model to conduct CoT with limited computation.

**Questions:**

Besides maths problems, does implicit chain-of-thought also work on other types of reasoning tasks? If not, what are the barriers that implicit CoT faces?

---

> ### Author Response · Authors · 2023-11-23
>
> Thank you for your comments.
>
> ### Re: it is hard to tell whether indeed the proposed system is conducting CoT reasoning or is simply learning some reasoning shortcuts
>
> Since the model trained for implicit CoT can do 5-by-5 multiplication, a task for which we don’t know of reasoning shortcuts, it is very likely that the model has learned to conduct reasoning internally.
>
> ### Re: The proposed method may not generalize compositionally to questions requiring more reasoning steps or just out-of-distribution data
>
> We are not making a claim that implicit CoT can generalize compositionally or to out-of-distribution data. Also, explicit CoT methods face similar issues (https://arxiv.org/abs/2305.18654). As one example, if we trained a transformer model on 4-by-4 multiplication, it is unlikely that it can do 5-by-5 multiplication. As another example, language models are known to not generalize to out-of-distribution data (https://openreview.net/forum?id=73OmmrCfSyy).
>
> ### Re: It forces the model to conduct CoT with limited computation
>
> We agree, and from empirical evidence it seems that implicit CoT needs a certain number of layers for tasks requiring more reasoning steps (such as GPT-2 Small on $5\times5$ multiplication). However, our method shows that there is untapped potential in large language models like GPT-4, which might have much more number layers than the number of reasoning steps in most tasks.
>
> ### Re: Does implicit chain-of-thought also work on other types of reasoning tasks
>
> We think it has the potential to do so, though demonstrating that is beyond the scope of this first paper.

---

### Official Review · Reviewer_kRur · 2023-11-01

**Soundness:** 2 fair
**Presentation:** 2 fair
**Contribution:** 2 fair
**Rating:** 3
**Confidence:** 4

**Summary:**

This paper proposes a new way to solve complex reasoning tasks (particularly, arithmetic reasoning tasks) without explicit chain-of-thought (CoT). to improve the efficiency of LLMs in reasoning. The authors propose a pipeline with 3 modules: a teacher model that encodes the chain-of-thoughts and provides representations for the CoTs; an emulator model that is trained to generate the encoding results from the teacher model during inference; and a student model that directly predicts the answer based on the encoding results from the teacher model/emulator. Small-scale empirical evaluations demonstrate the potential of the proposed method, which achieves similar performance with CoT prompting and higher efficiency.

**Strengths:**

The idea is straightforward and the motivation is clear.

**Weaknesses:**

1. **The writing is not clear and some paragraphs are not rigorous enough.** The proposed pipeline includes 3 different modules, and the explanations of how they work are hard to follow. In the `information Extraction` paragraph on page 5, the representation of the CoTs is extracted from the diagonal elements of the matrix $z$. However, matrix $z$ is often not a square matrix. In that case, what does it mean to extract the elements from $z_{11}$ to $z_{LL}$? Does it mean that the hidden states of tokens after the position $T$ are discarded when $T<L$? Also, how to process the case when $L<T$?

2. **Missing an important baseline.** The proposed method can be regarded as **compressing CoTs into a vector**. There could be multiple ways to compress CoTs into a single vector and then train another model to generate the compressed representations for CoTs during the inference process. Why it is the best choice to extract the intermediate states of the teacher model as the compression results? From my perspective, a more elegant way to compress CoTs should be to train an auto-encoder to map the original CoTs into a single vector. The auto-encoder ensures that the compressed vector contains all the necessary information to recover the CoTs and can be used to predict the answer directly. Specifically, denoting the teacher model as $f_{T}(\cdot; \theta_{T})$ with parameter $\theta_{T}$, the auto-encoder model as $g_{enc}(\cdot;\phi_{enc}) and g_{dec}(\cdot;\phi_{dec})$ with parameter $\phi$, the student model as $f_{s}(\cdot; \theta_{s})$. We extract the CoTs from the teacher model given a question $q$ as $c = f_{T}(q;\theta_{T})$ where $c = (c_{1}, \dots, c_{N})$ is the CoT. Then the auto-encoder is trained with the objective

   $$\max _{\phi} P(\hat{c _ i}|z, \hat{c} _ {<i}), \text{where} z=g(c;\phi _ {enc}) \text{and} P(\hat{c _ i}|z, \hat{c} _ {<i}) = g _ {dec}(z, \hat{c} _ {<i};\phi _ {dnc})$$

   Then the student model is trained to maximize the generation probability of the target answer conditioned on the question and $z$. The proposed method is exactly a special case of the above framework, where the auto-encoder is replaced by the proposed information extraction method. The author should demonstrate why the proposed method outperforms the general framework above, or why we should choose the proposed design. Otherwise, the method is too intuitive.

3. **Insufficient empirical evaluation.** The authors only verify that implicit CoT can boost performance. However, it is still not comparable with standard CoTs. Also, the efficiency improvement is less significant (or necessary) considering the poor performance of implicit CoTs. Finally, I would like to see further discussions on why should we consider implicit CoTs besides the reason for efficiency, especially considering the significantly increased training cost and data collection cost.

**Questions:**

Please refer to the weakness above. Although I give a low rating to this paper, I would be delighted to increase my rating given the questions addressed.

---

> ### Author Response · Authors · 2023-11-23
>
> Thank you for your comments.
>
> ### Re: matrix $z$ is often not a square matrix. In that case, what does it mean to extract the elements from $z_{11}$ to $z_{LL}$?
>
> We presented how to deal with the case of (almost certainly) non-square matrices in the paragraph next to “Information Extraction” (page 5 - Variable Length Chain-of-Thought). In short, we introduced a hyperparameter to specify the distance between the columns taken, with two variants, one of which used a variable spacing which is determined by the ratio of reasoning steps to number of layers. In this variant, when there are more tokens than layers ($T>L$), the spacing will be greater than 1; when there are more layers than tokens ($T<L$), the spacing will be less than 1 (the spacing is a float number but the actual position taken is rounded to an integer). In both cases, the last vector is taken from the last column of the teacher's hidden states.
>
> ### Re: Missing an important baseline
>
> We agree that the baseline you mentioned is more powerful than our current information extraction method. However, we wanted to show that even with the simple method we can realize the concept of implicit chain of thought. We consider our method as one step towards implicit chain-of-thought and we believe there are a lot of improvements that can be made on top of our current method, including the one that the reviewer posits.
>
> That said, we have conducted an experiment on 5X5 multiplication following your suggestions. In particular, we view the teacher states as a sequence of column-wise vectors $c_1, c_2,\cdots c_T$ following your notation. For the encoder, we used a BERT followed by mean pooling and layer normalization (we replaced the embedding layer with a linear projection layer to make shapes match). For the decoder, we used a GPT-2 model (again, we replaced the embedding layer with a linear projection layer to make shapes match). The auto-encoding objective is to predict the teacher hidden hidden states at the next timestamp, conditioned on past teacher hidden states and the encoded teacher states, where the conditioning is implemented via addition to all input hidden states. We then use the encoded teacher states to replace the extracted diagonal elements for both training the mind-reading student and the emulator. We conducted experiments on $5\times5$ Mult and found that GPT-2 Small got 0.2% accuracy and GPT-2 Medium got 0.7% accuracy even using oracle teacher states (code can be found at https://anonymous.4open.science/r/implicit_chain_of_thought/src_autoencoder/train_mind_reading_student.py). That said, this doesn’t mean that this approach is worse — there’s plenty of room for architectural optimization and hyperparameter search which we didn’t have time to exhaust. We consider this approach a potentially improved version of our current implementation, which is only a preliminary step towards realizing implicit chain-of-thought.
>
> ### Re: Not comparable to explicit CoT
>
> We agree that our method has lower accuracy than explicit CoT. Besides the higher inference throughput, our deeper aim is to break from the notion that LMs must adhere to human-like reasoning. Instead, we hope to enable LMs to develop their own reasoning pathways, leveraging their capabilities quite different from those we provide them with. There could also be interesting follow-up work from ours, such as a hybrid mode of explicit/implicit reasoning, where LMs could use implicit CoT for quick responses when confident, and switch to explicit CoT for challenging queries, or even use the internal reasoning to solve multiple problems at once (similar to the data multiplexing task in https://arxiv.org/abs/2302.12441).

---

### Meta-Review · Area_Chair_AHx9 · 2023-12-11

**Metareview:**

The paper develops an implicit chain-of-thought method, which seeks to replace the explicit natural language reasoning steps (“horizontally”) with implicit reasoning steps in hidden states (“vertically”). It shows some promising results on arithmetic tasks and GSM8K. However, there are several major weaknesses pointed out by the reviewers, such as unclear presentation, weaker interpretability, and unconvincing experimental results. In particular, the results of the proposed method is still significantly falling behind the explicit CoT method. And such implicit “vertical” CoT method may be significantly restricted by the number of layers in the LLM, making it fundamentally restricted on problems that requires long reasoning steps.

**Justification For Why Not Higher Score:**

This paper has many major limitations as pointed out by reviewers (and summarized in the above meta review).

**Justification For Why Not Lower Score:**

N/A

---

### Decision · Program_Chairs · 2024-01-16

Reject